# Relationship between Health Literacy and Knowledge, Compliance with Bowel Preparation, and Bowel Cleanliness in Older Patients Undergoing Colonoscopy

**DOI:** 10.3390/ijerph19052676

**Published:** 2022-02-25

**Authors:** Minju Gwag, Jaeyong Yoo

**Affiliations:** 1Graduate School of Nursing, Chosun University and Endoscopy Center, Gwangju Hyundai Hospital, Gwangju 61452, Korea; kmj79eh78@daum.net; 2Department of Nursing, College of Medicine, Chosun University, Gwangju 61452, Korea

**Keywords:** bowel preparation, colonoscopy, older people, health literacy

## Abstract

Compared to young adults, it is difficult for the older people with relatively low health literacy to perform proper bowel preparation for a colonoscopy. This study aims to identify the relationship between knowledge, compliance with bowel preparation, and bowel cleanliness with health literacy in older patients undergoing colonoscopy. The participants were 110 older people undergoing colonoscopy, recruited from an endoscopy hospital in G metropolitan city, South Korea. Data obtained from a structured questionnaire that included items on health literacy and knowledge of and compliance with bowel preparation, and the Aronchick bowel cleanliness scale. The data were analyzed using descriptive statistics, χ-test, Pearson’s correlation, *t*-test, and ANCOVA. Participants who were younger and those with a higher education level and better economic status had a statistically significantly higher health literacy level. Older people with a health literacy level of 7 points and above had a higher knowledge level and bowel cleanliness index, a showed better compliance with bowel preparation. The results highlight the need for developing a customized education intervention program that can improve health literacy for successful bowel preparation and examination of the older population undergoing colonoscopy.

## 1. Introduction

According to the “Causes of Death Statistics in 2020” from the Korea National Statistical Office, 27.0% of all deaths was attributable to cancer, accounting for a total of 82,204 people, making it the number one cause of death among Koreans [1]. Colorectal cancer is the third most common type of cancer, accounting for 11% of all cancer-related deaths [1], and its prevalence rate is the second highest in the group of older people aged 65 years and above. The American Society for Gastrointestinal Endoscopy guidelines particularly consider people in this age group to be at high-risk for colorectal cancer [2]. In Korea, this risk has been increasing every year [1]. Although colorectal cancer can be treated if detected early, it often progresses without symptoms, making early detection difficult [2,3].

Colonoscopy is considered the most useful diagnostic test for early detection of colorectal cancer and various other colon lesions [2]. Its diagnostic efficiency is highly dependent on the quality of bowel preparation [4,5]. Inadequate bowel preparation may result in residual feces masking clinically important lesions, leading to major tumors going undetected [6,7,8]. As a result, the colonoscopy execution time may be prolonged, or the examination may not proceed as it should [5,9]. When colonoscopy is delayed or repeated, the accuracy of the procedure and patient satisfaction decreases, and additional medical costs are incurred [7,10,11]. Therefore, bowel preparation is an important colonoscopy quality management index, and it is necessary to record bowel cleanliness during colonoscopy and continuously monitor patient outcomes [5,12].

The guidelines for quality control indicators for colonoscopy presented by the US Multi-Society Task Force on Colorectal Cancer emphasize that adequate bowel preparation should be part of at least 85% of the total number of colonoscopy cases [2]. However, in clinical practice, about 20–25% of colonoscopies are reportedly performed without adequate bowel preparation [4]. Unlike other diagnostic tests that do not have pre-preparation procedures or are relatively simple, colonoscopy requires the examinee to fully understand the preparations before the colonoscopy and to perform complex preparation procedures [13]. However, it is difficult for older patients with low-comprehension of medical information or a low education level to accurately understand and implement the guidelines for bowel preparation [11,14,15]. For successful colonoscopy in older people, it is necessary for them to fully understand the need for bowel cleansing and the preparation process before examination. Thus, it is important for older patients undergoing colonoscopy to have high health literacy [15,16,17].

In contrast to literacy, which refers only to the ability to read and write words or sentences, health literacy is defined as “the degree to which individuals have the capacity to obtain, process, and understand basic health information and services needed to make appropriate health decisions [18].” In previous studies on health literacy and bowel preparation [10,11,14], examinees with lower health literacy had lower education levels and lacked knowledge about bowel preparation. Furthermore, their level of compliance with the pre-colonoscopy guidelines was low, and inadequate bowel preparation was observed more frequently [11,19]. Therefore, postponement of colonoscopy leads to a wait of on average 14.1 months until re-examination, with the risk of missing colon adenomas and their possible worsening into advanced colorectal cancer [20]. The health literacy of patients undergoing colonoscopy greatly influences the level of bowel preparation and, the prognosis of colorectal disease.

In Korea, it has been reported that the proportion of people with low health literacy is 38%, which is higher than the average of 22% in OECD countries [21]. According to a survey of older people by the Korea Institute for Health and Social Affairs, 31.6% of the older people over 65-years-of-age had an education level below middle school, and about 10% reported difficulty in deciphering text [22]. According to the guidelines of the American Gastroenterological Association, the effect of bowel cleansing in older adults may be lower than that in young adults when the same guidelines for bowel preparations are applied because colonic motility is lower in older adults [23]. However, at present, most hospitals in Korea do not consider the characteristics of older people; instead, they provide a paper-based guide with instructions for bowel preparation, and then explain it verbally only once, making it difficult to deliver sufficient information [24,25].

As the older population is expected to increase, it is important to identify their health literacy level and provide an educational intervention program to improve it [26,27]. According to recent studies on the relationship between health literacy and healthcare of older people, the emphasis is on providing customized educational intervention programs by considering older adults’ existing level of health literacy [28,29]. Nevertheless, studies on health literacy in older Korean individuals aged over 65 years of age undergoing colonoscopy are rare. In particular, studies on the relationship between health literacy and knowledge, compliance with bowel preparation, and bowel cleanliness in older people have not yet been conducted. Through this study, we intend to provide basic data for developing a customized nursing education intervention program that can improve health literacy regarding proper bowel preparation and successful colonoscopy for older people.

This study was conducted to identify relationships between health literacy and knowledge, compliance with bowel preparation, and bowel cleanliness in elderly patients who are undergoing colonoscopy. To elaborate, we aimed to identify the following:The participants’ general characteristics, knowledge, compliance of bowel preparation, bowel cleanliness, and health literacy level;Differences in health literacy, knowledge, compliance of bowel preparation, and bowel cleanliness based on participants’ characteristics;The correlation between health literacy and related variables;The differences in the knowledge, bowel preparation compliance, and bowel cleanliness based on participants’ health literacy level.

## 2. Materials and Methods

### 2.1. Study Design and Participants

This study used a cross-sectional descriptive design. Using a convenience sampling method, older patients aged 65-years and older were recruited in October 2020. Participants were patients who visited H Hospital with 275 beds, located in G Metropolitan City, Korea. The participants included older adults who did not have any problems with cognitive ability, communication, and daily life activities. The evaluation of the participants’ cognitive level, communication and daily living level was confirmed through the initial interview with the gastroenterologist at the outpatient colonoscopy room. Only participants who fully understood the purpose and process of the study and voluntarily consented were included. Further, 95 samples were calculated as the minimum sample size when the effect size was 0.5, with a significance level of 0.05 and a power of 0.80; the samples were set in a two-sided test using the G-power 3.1 program. Considering the dropout rate of 15%, 114 older individuals were recruited. After excluding four individuals whose responses were incomplete or invalid, data of 110 individuals were analyzed.

### 2.2. Measurements

This study used a structured questionnaire consisting of 49 items: 14 items on the participants’ general characteristics, 12 items on health literacy, 11 items on knowledge, 11 items on compliance with bowel preparation, and 1 item on bowel cleanliness. The tools used in this study were obtained with the permission of the original authors.

***Health literacy***. Health literacy measures have been developed based on the Rapid Estimate of Adult Literacy in Medicine (REALM) tool that assesses adult patients’ understanding of medical words and related terms [18]. In this study, health literacy was measured using the short form of the Korean Health Literacy Scale for the Elderly (S-KHLS), which consists of 12 items to measure older individuals’ ability to understand medical information [30]. This short form was narrowed down to 12 items from the original 24 items developed by Lee and Kang [31]. First, in the health-related terms section (five items), questions pertained to whether people understood the meanings of alcohol consumption, obesity, and lifestyle-related diseases. The comprehension and numeracy sections (seven items) included basic-dose calculation, checking the date of the appointment card, reading the medication guide, and reading the nutritional information table. A correct answer received 1 point, an incorrect answer received 0 points, and possible scores ranged from 0 to 12 points. A higher score indicated a higher degree of health literacy. Clinically, a total score of more than 7 is considered high health literacy, and a score of less than 7 is classified as low health literacy [30]. Cronbach’s α was 0.80 in the original study and 0.77 in the current study.

***Knowledge of bowel preparation***. Knowledge of bowel preparation was measured using a tool developed by Yu [32], which was later revised and supplemented by Cho and Kim [33] based on the guidelines of the Korean Society of Gastrointestinal Endoscopy. The knowledge tool is largely divided into two sub-domains: information on bowel cleansing agents and dietary knowledge prior to colonoscopy. Knowledge of bowel cleansing agents consists of five items related to the dosage, administration, and storage method of the agents. The contents of the six questions about dietary knowledge consist of the type of diet to be restricted and the meal time the day before colonoscopy. A correct answer is worth 1 point and an incorrect answer and “do not know” are worth 0 points; the higher the score, the higher the knowledge level. In Cho and Kim’s study [33], KR-20 was 0.81 and 0.84, respectively, and in this study, it was 0.71 and 0.88, respectively.

***Compliance of bowel preparation***. Compliance with bowel preparation was measured using a tool developed by Yu [32], and later revised and supplemented by Cho and Kim [33] based on the guidelines of the Korean Society of Gastrointestinal Endoscopy. It consisted of five questions on how well the participants followed the guidelines for taking bowel-cleansing agents and six questions on compliance with the dietary guidelines. Responses were rated on a four-point Likert scale, with 1 = “not at all”, 2 = “a little bit”, 3 = “most of the time”, and 4 = “very well”. Higher scores indicated better compliance with the guidelines. In Cho and Kim‘s study [33], Cronbach’s α was 0.68 and 0.81, respectively, and in this study, Cronbach’s α was 0.72 and 0.81, respectively.

***Bowel cleanliness***. The Aronchick Bowel Preparation Scale, which has been shown to have good validity, was used to evaluate the degree to which the colonic mucosa can be clearly observed by removing feces from the large intestine [34]. The doctor who performs the colonoscopy evaluates the amount of feces remaining throughout the colon after the colonoscopy is performed. The degree of bowel cleanliness is rated as 1 = “excellent”, 2 = “good”, 3 = “fair”, 4 = “poor”, or 5 = “inadequate”. When no solid stool is observed during colonoscopy and only a small amount of liquid that can be easily removed by suction is observed, it is evaluated as “excellent”. When colonoscopy is impossible owing to a full solid rectal stool, it is rated as “inadequate”.

***General characteristics of the participants***. The general characteristics of the participans included 14 items: gender, age, religion, marital status, education level, income, preferred eating habits, colonoscopy experience, abdominal surgery experience, underlying diseases, family history, family support, constipation, and types of health insurance.

### 2.3. Data Collection

Data collection was performed in outpatient department of the colonoscopy laboratory of H Hospital, located in G Metropolitan City. The research team explained the overall study in detail to the colonoscopy doctor and department head of the endoscopy center before seeking approval and cooperation related to data collection. Because the hospital where this study was conducted has an endoscopy center, the number of daily visits of adult patients is high. Among the adults visiting the endoscopy center, the older people who met the selection criteria were continuously and conveniently recruited. Study participants were asked to fill out the form 30 min before the start of the colonoscopy. It took about 15–20 min to fill out the questionnaire. The questionnaire was completed in a separate area in the outpatient setting where privacy was maintained. Participants were given time to fully understand and the contents of the questionnaire, and they were allowed to ask the researcher questions. The bowel cleanliness index after colonoscopy was directly written on the case report form by the gastroenterologist who performed the colonoscopy. 

### 2.4. Ethical Considerations

Prior to data collection, this study was reviewed and approved by the Institutional Research Ethics Review Board of University C (Approval number: 2-1041055-AB-N-01201825). The purpose of the study and the method of participation were sufficiently informed to older people aged ≥ 65 years who visited the colonoscopy center if they met the criteria for participant selection. It was explained that the collected data would remain anonymous and would be used only for research purposes. Written consent was also obtained from the study participants. The participants were informed that they could withdraw from the study at any point of time. To ensure the anonymity of the participants, the surveys were coded and collected without personal information. After the survey was completed, a small gift (about USD 3) was provided to the study participants. None of the participants complained of any unexpected discomfort when participating in the survey. 

### 2.5. Data Analysis

Data were analyzed using IBM SPSS v 26.0. The levels of health literacy, knowledge, implementation, and intestinal cleanliness were analyzed using descriptive statistics. Differences in study variables according to participant characteristics were analyzed using *t*-tests, analysis of variance (ANOVA), and Scheffé post hoc tests. Correlations between variables were analyzed using Pearson’s correlation. The *t*-test and analysis of covariance (ANCOVA) were used to determine the differences in the relevant variables according to health literacy.

## 3. Results

### 3.1. General Characteristics of the Study Participants

Participants’ mean age was 71.01 ± 5.05 years, 63.6% were women, and 77.3% reported that they were living with a spouse. Furthermore, 56.4% had graduated from middle school or lower; 30.0% earned a current monthly income; and most of them had health insurance. The proportion of participants who received family support while preparing for colonoscopy was 52.7%, while the remaining answered that they had prepared without assistance. Regarding health-related characteristics, 54.5% had an underlying disease and 67.3% had a previous colonoscopy experience. About 30.0% of the participants usually preferred meat in their diet and 25.5% of subjects were experiencing constipation problems at the time of the survey (Table 1).

### 3.2. Level of Health Literacy, Knowledge, Compliance, and Bowel Cleanliness

The health literacy score was 7.55 ± 2.95 out of 12 points. Among the sub-domains, the health-related terms section had a score of 4.02 ± 1.20 points, while comprehension and numeracy had a score of 3.54 ± 2.04 points. The knowledge of bowel preparation score was 7.97 ± 2.81 out of 11 points. The average score for knowledge about bowel cleansing agents was 2.96 ± 1.50, and that for dietary knowledge was 5.01 ± 1.77.

The average level of compliance with the bowel preparation guidelines was 3.38 ± 0.38 on a four-point scale. The level of taking the bowel cleansing agents was at 3.25 ± 0.46 points, and compliance with the dietary guidelines was at 3.52 ± 0.58 points. The average bowel cleanliness score was 3.32 ± 1.15, that is, 17.3% for “excellent”, 25.5% for “good”, 10.0% for “poor”, and 9.1% for “inadequate”, based on the Aronchick Bowel Preparation Scale (Table 2).

### 3.3. Differences in Health Literacy, Knowledge, Compliance, and Bowel Cleanliness according to Participants’ Characteristics

Health literacy showed statistically significant differences for age (F = 18.44), education level (t = −4.16), monthly income (t = −3.88), insurance type (t = 2.03), and constipation (t = −2.31). The level of knowledge was statistically significantly different according to age (F = 4.01) and family support (t = −2.64); the level of compliance with the guidelines was statistically significantly different by gender (t = 2.45), marital status (t = 2.29), monthly income (t = 2.19), and family support (t = −2.14). There was a statistically significant difference in bowel cleanliness according to monthly income (t = 2.50) (Table 3).

### 3.4. Correlations among Health Literacy, Knowledge, Compliance, and Bowel Cleanliness

On the bowel cleanliness scale, a score of 1 out of 5 indicated the best condition, based on the Aronchick Bowel Preparation Scale. To facilitate the interpretation of the results of the correlation analysis, the results were analyzed using inverse transformation. Health literacy had a statistically significant positive correlation with knowledge (r = 0.20), compliance (r = 0.29), and bowel cleanliness (r = 0.36). The higher the health literacy, the better the knowledge, compliance with guidelines, and bowel cleanliness. In the sub-domain analysis, knowledge and compliance related to taking bowel cleansing agents were, statistically, significantly related to health literacy; however, knowledge and compliance level related to the diet before the test were not correlated (Table 4).

### 3.5. Differences in Knowledge, Compliance and Bowel Cleanliness by Health Literacy

Older people with a health literacy level greater than seven points had a high level of knowledge (taking a bowel cleansing agent), a high level of compliance for bowel preparation, and a high bowel cleanliness index (all *p* < 0.05; Table 5).

## 4. Discussion

Recently, the World Health Organization declared health literacy as one of the main strategies for disease prevention and health promotion through the Shanghai Declaration; it emphasized the need for greater attention toward health literacy by healthcare providers as well [35]. Therefore, in this study, the relationships between health literacy and knowledge, compliance with bowel preparation, and bowel cleanliness, and the differences among these variables based on health literacy, were identified in older Koreans who had undergone colonoscopy.

In this study, the health literacy level of older patients who had previously undergone a colonoscopy was scored at 7.55 + 2.95 out of 12 points, much lower than the average score of 8.21 in a study of 239 older people with coronary artery disease [36] and 10.52 in a study of 315 adults with community-dwelling hypertension using the same measurement tool [37]. However, it was higher than the average score of 6.08 in a study of 134 elderly people with cardiovascular disease living in rural areas of Korea [38]. Most previous studies report that approximately 40% of older people have low health literacy, a conclusion that is similar to ours [39,40]. There is criticism, however, that studies showing higher literacy often include adults aged 19-years or older or those older adults who regularly visit hospitals because of disease comorbidity [37,38]. It can also be said that the low health literacy of the rural older people, who live farther from hospitals, could mean that literacy depends on residential characteristics. In general, the lower health literacy among older people suggests they may lack the ability to successfully perform bowel preparation before a colonoscopy as per the guidelines [14,23,26]. In such cases, we expect a delay in colonoscopy examination time, along with more test-related side effects, a higher retest rate, and additional medical costs.

Regarding participants’ characteristics, we found that older patients and those with poor education or low income had lower health literacy, similar to prior studies [11,19,41,42]. However, there was no gender difference in health literacy, although a prior study showed comparatively lower literacy among female patients with cardiovascular disease [38] or undergoing hemodialysis [43]. Nevertheless, there are no conclusively found gender differences across the literature [10,44,45]. Thus, our results need to be reinterpreted by considering the significant differences in educational and socioeconomic levels between the genders; that is, the level of health literacy may differ depending on socioeconomic level, even for older people of similar ages. It should be noted that we found compliance with bowel preparation guidelines to be higher in men and those currently living with a spouse. 

Although there was no statistically significant difference in this study as well, it was found that the health literacy, knowledge, and compliance levels of the older female were generally low, so it is necessary to pay attention to these population groups. Since older women living alone have low health literacy or are likely to perform bowel preparations inappropriately, it is necessary to focus on education and economic level along with health literacy in the initial assessment stage when older women visit the hospital. In addition, the development of educational interventions reflecting the physical and psychological support of older women with various health conditions, such as menopause, osteoporosis, thyroid disease, and abdominal obesity, should be considered. Therefore, rather than a single characteristic of gender, the economic/education level, presence or absence of supportive resources, such as a spouse or family, health condition, and cultural differences, may have a complex effect on health literacy [10,31,46,47]. Follow-up studies that reflect these differences are presumed to be necessary.

The average level of knowledge related to bowel preparation of the participants was 7.97 ± 2.81 points (72.4 points out of 100). In the sub-domain, the score for knowledge of bowel cleansing was 59.0 and for knowledge related to diet was 83.5. In a previous study of 98 adults who had undergone colonoscopy [48], these scores were 59 and 61, respectively. We can conclude then that the older participants in our study had higher dietary knowledge. Thus, the higher the health literacy, the more statistically significant was the knowledge of bowel preparation. This means that the level of knowledge about bowel preparation may vary according to health literacy. In a study of elderly patients with heart failure [43], those with high health literacy had a higher-level of disease-related knowledge; meanwhile, a study of hypertensive patients living in the community [37] showed a statistically significant positive correlation between health literacy and knowledge level. This suggests that individualized health education should be provided to improve the knowledge of older people by identifying their health literacy when they first visit the hospital.

We recommend a critical review of the education on colonoscopy preparation that has been provided by hospitals. This way, experts can replace difficult terms with easy-to-understand ones for older people. Other graphic changes can also be made, such as increasing the font size or inserting additional pictures or illustrations for better comprehension [15,46,49,50]. If educational materials are made into videos, patients should be provided with easy-to-access QR codes or URL addresses for repeated viewing at home. Providing an intervention that considers the health literacy of older patients can improve their understanding of the entire colonoscopy process, increase adherence to preparation guidelines, and, thus, lead to more successful colonoscopies. However, in this study, we found no difference in knowledge about diet for intestinal preparation. We, thus, recommend in-depth initial assessment and interviews relating to meal preparation during this stage, that is, checking whether patients depend on their spouse or themselves, or continue their usual eating habits.

The average level of compliance for bowel preparation was 3.38 ± 0.38 points (84.5 points out of 100). In the sub-domain, the score for compliance with bowel cleansing agents was 81.2 and for compliance with dietary guidelines was 88.0. In a previous experimental study on Korean adults [48], compliance with bowel cleansing agents was reported to be 74.1 points, and compliance with dietary guidelines was reported to be 70.4 points—both lower than in our study. The compliance in the group with high health literacy was 3.47 ± 0.42, and in the group with low health literacy it was 3.27 ± 0.28. Nguyen et al. [11] reported that 86.7% of patients with inadequate bowel preparation did not follow the guidelines provided by the medical staff. Compliance with dietary guidelines was reported to be relatively higher than the agent intake, possibly because the data collection method relied on self-reporting. Many older adults are also reluctant to admit that they have failed to comply with the guidelines or were unaware of their non-compliance. Therefore, a review of lifestyle, including eating habits, is necessary before pre-colonoscopy education is provided; we also recommend education that is both customized and differentiated by lifestyle. Another fruitful approach would be to conduct an observational evaluation by a spouse or caregiver in parallel with self-reports when collecting data. In particular, in the case of the older people living alone because of divorce or the death of a spouse, or those who are underprivileged and receive little to no social support, it is more important to evaluate whether they can follow the provided guidelines.

The bowel cleanliness level was an average of 3.32 ± 1.15 points on the Aronchick scale [34], and 42.8% were rated “excellent” or “good”. The group with high health literacy scored 2.39 ± 1.22 and the one with low health literacy scored 3.06 ± 0.93, indicating that, bowel cleanliness was significantly better among those with high health literacy. In an educational intervention study targeting 72 older people, based on mobile text messages and counseling [51], the experimental group scored 1.87 and the control group scored 2.75. In an experimental study using the same measurement tool [52], an 8-min training video improved the bowel cleanliness of the experimental group by 32% compared to the control group. In previous studies, age over 60-years [10,47], low educational level and health literacy [26], and low socioeconomic level [50] were suggested as predictors of inadequate bowel cleansing. Therefore, the medical staff performing a colonoscopy should pay special attention to older patients with these risk factors.

We also found that the higher the health literacy of the older people, the higher the knowledge of bowel preparation, level of compliance with bowel-cleansing agents, and index of bowel cleanliness. For older people to successfully carry out the guidelines for bowel preparation, systematic interventions to increase their health literacy should be conducted to improve their understanding and knowledge of the procedure. A recent review study on educational interventions for bowel preparation reported that interventions, such as educational booklets, visual materials using illustrations, videos, short messages, phone counseling, social network services, and other applications were effective [46]. In a study of 256 patients, educational interventions to improve health literacy improved bowel preparation indicators [53], similar to another application-based intervention study of 160 patients [54]. In an intervention study of 770 participants using the social networking service platform WeChat, bowel preparation indicators and adenoma detection rates also improved [55]. It is also necessary to develop and provide a newly configured internet and social media-based information platform for older people to use. In addition, it can be a good idea to use social media platforms with the highest access volume in each country or platform services that older adults are already familiar with how to use them among older people [56].

When providing education for older people to prepare for colonoscopy, it is necessary to consider what educational approach should be taken to effectively convey knowledge and help patients prepare and implement the guidelines on their own. In particular, unlike other diagnostic tests, the preparation process for colonoscopy requires older people to perform it themselves. Therefore, differentiated education is necessary whereby individual patients’ health literacy is taken into consideration [15], especially that of older patients making their first visit to the hospital. For older patients with low health literacy, simulation-based education in which patients can directly participate and interact may be useful, along with interactive audio–visual materials that can be repeatedly accessed at any time and place [33,48]. Educators may also consider technological interventions, such as augmented reality and virtual reality, to create an immersive learning-experience for older patients [27,57]. Because such patients may also have weak cognitive abilities [39], ensuring that they have retained all the information is necessary through the teach-back method, which has been known to be effective for this age group [58]. In this method, the learner is made to repeat what they have understood to the educator, that is, in our case, the medical staff [58,59]. The Heart Failure Society of America recommends actively carrying out patient-centered education that considers the patient’s health literacy level, including teach-back education [60]. Once the communication process improved, patients’ health literacy level was recorded in the medical register so that the medical staff could refer to it before providing any intervention [60]. In this study, a 12-item shortened tool was used to measure the health literacy of older patients who visited the hospital for colonoscopy. If it is actively used in the clinical field, it will contribute to the delivery of accurate health information to older people and improve health outcomes.

We believe that our study is meaningful because it investigates the health literacy, knowledge, implementation of guidelines, and bowel cleanliness of older Korean patients undergoing colonoscopy. Our results suggest the need for more and better education on colonoscopy preparation; it must also be ensured that the education is differentiated with respect to the health literacy of each patient. We hope that modifying current approaches accordingly may increase the quality and success of this procedure, and ultimately improve patient outcomes.

Nevertheless, our study has some limitations. Because we employed a small number of older Korean people at one hospital as our participants, our results may have limited generalizability. One solution is expanding the sample size to include more patients and institutions across a wider geography. In addition, in a recent systematic review [10] that analyzed 24 studies with 49,868 patients, the types of major diseases (e.g., diabetes, hypertension, liver cirrhosis, and stroke) and the use of antidepressants and opioid analgesics have also been reported to be associated with inadequate bowel preparation. Therefore, future studies on adherence to guidelines for medical procedures should take into account the comorbidities and medication-related characteristics of older people.

## 5. Conclusions

For a successful colonoscopy through proper bowel preparation, the health literacy of older people should be considered. We, thus, recommend a critical review of the existing colonoscopy education provided to older people at hospitals. For older peole who are scheduled for a colonoscopy, health literacy should be evaluated in advance, and individualized education should be planned and provided. Regarding future research, we propose an experimental study that applies an individual educational intervention program to improve the health literacy of older people and evaluates the improvement of various patient outcomes, such as the incidence of side effects, frequency of retests, and reduction in medical costs.

## Figures and Tables

**Table 1 ijerph-19-02676-t001:** General characteristics of study participants (*n* = 110).

Variables	(Mean ± SD)	*n* (%)	Variables		*n* (%)
Gender	Male	40(36.4)	Underlying disease	Yes	60(54.5)
	Female	70(63.6)		No	50(45.5)
Age	65–69	51(46.4)	Family history	Yes	10(9.1)
(71.01 ± 5.05)	70–74	37(33.6)		No	100(90.9)
	≥75	22(20.0)			
Marital status	Married	85(77.3)	Abdominal surgery	Yes	22(20.0)
	Unmarried	1(0.9)	experience	No	88(80.0)
	Divorced	4(3.6)	Colonoscopy experience	Yes	74(67.3)
	Bereavement	20(18.2)		No	36(32.7)
Education level	Elementary	31(28.2)	Meat preference	Preferred	33(30.0)
	Middle	31(28.2)		Moderate	69(62.7)
	High	38(34.5)		Not preferred	8(7.3)
	Bachelor	9(8.2)			
	Graduate	1(0.9)	Having constipation	Yes	28(25.5)
				No	82(74.5)
Religion	Christianity	15(13.6)			
	Catholicism	10(9.1)	Family support in	Yes	58(52.7)
	Buddhism	17(15.5)	Preparing for colonoscopy	No	52(47.3)
	No religion	68(61.8)			
Monthly income	Yes	33(30.0)	Health insurance	Insurance	101(91.8)
	No	77(70.0)		Assistance	9(8.2)

**Table 2 ijerph-19-02676-t002:** Level of health literacy, knowledge, compliance, and bowel cleanliness (*n* = 110).

Variables	Categories	*n*(%)	Mean ± SD	Min/Max/Range
Health literacy	Total mean		7.55 ± 2.95	1/12.0/0–12
	Sub 1: Health-related terms		4.02 ± 1.20	1/5.0/0–5
	Sub 2: Comprehension and numeracy		3.54 ± 2.04	0/7.0/0–7
Knowledge of bowel preparation	Total mean		7.97 ± 2.81	0/11.0/0–11
	Sub 1: Knowledge about taking bowel-cleansing agents		2.96 ± 1.50	0/5.0/0–5
	Sub 2: Dietary knowledge for colonoscopy		5.01 ± 1.77	0/6.0/0–6
Compliance of bowel preparation	Total mean		3.38 ± 0.38	2.15/4.0/1–4
	Sub 1: Taking bowel-cleansing agents		3.25 ± 0.46	1.6/4.0/1–4
	Sub 2: Dietary guidelines for colonoscopy		3.52 ± 0.58	1.5/4.0/1–4
Bowel cleanliness	Total mean		3.32 ± 1.15	1.0/5.0/1–5
(Aronchick scale)	Excellent (1 point)	19(17.3)		
	Good (2 points)	28(25.5)		
	Fair (3 points)	42(38.2)		
	Poor (4 points)	11(10.0)		
	Inadequate (5 points)	10(9.1)		

**Table 3 ijerph-19-02676-t003:** Differences in health literacy, knowledge, compliance, and bowel cleanliness by participants’ characteristics (*n* = 110).

Variables	Categories	*n* (%)	Health Literacy	t or F(*p*)	Knowledge	t or F(*p*)	Compliance	t or F(*p*)	BowelCleanliness	t or F(*p*)
			Mean ± SD		Mean ± SD		Mean ± SD		Mean ± SD	
Gender	Male	40(36.4)	8.15 ± 2.67	1.61 (0.110)	8.33 ± 2.56	0.99 (0.323)	3.50 ± 0.31	2.45 (0.016)	3.33 ± 1.16	−0.47 (0.963)
	Female	70(63.6)	7.21 ± 3.06		7.77 ± 2.94		3.32 ± 0.40		3.31 ± 1.15	
Age	65–69 ^a^	51(46.4)	9.08 ± 2.33	18.44 (<0.001) a > b > c	7.86 ± 3.01	4.01 (0.021) b > c	3.41 ± 0.42	0.71 (0.493)	3.55 ± 1.19	2.88 (0.061)
	70–74 ^b^	37(33.6)	6.84 ± 2.61		8.84 ± 2.17		3.40 ± 0.30		3.27 ± 0.93	
	≥75 ^c^	22(20.0)	5.23 ± 2.88		6.77 ± 2.91		3.30 ± 0.38		2.86 ± 1.28	
Marital status	Married	85(77.3)	7.84 ± 2.77	1.86 (0.065)	7.99 ± 2.69	0.11 (0.916)	3.43 ± 0.36	2.29 (0.024)	3.28 ± 1.14	0.60 (0.549)
	Others	25(22.7)	6.60 ± 3.38		7.92 ± 3.24		3.23 ± 0.44		3.44 ± 1.19	
Education Level	≤middle school	62(56.4)	6.55 ± 2.89	−4.16 (<0.001)	7.70 ± 3.07	−1.12 (0.266)	3.34 ± 0.35	−1.44 (0.152)	3.26 ± 1.07	−0.62 (0.535)
	≥high school	48(43.6)	8.85 ± 2.50		8.31 ± 2.42		3.44 ± 0.41		3.40 ± 1.25	
Religion	Yes	42(38.2)	7.80 ± 2.91	0.711 (0.478)	8.09 ± 2.51	0.358 (0.721)	3.32 ± 0.42	−1.24 (0.216)	3.16(1.22)	1.088 (0.279)
	No	68(61.8)	7.39 ± 2.97		7.89 ± 2.99		3.41 ± 0.34		3.41(1.09)	
Monthly income	Yes	33(30.0)	9.18 ± 2.44	−3.88 (<0.001)	7.82 ± 3.09	−0.38 (0.708)	3.50 ± 0.38	2.19 (0.031)	3.73 ± 1.15	2.50 (0.014)
	No	77(70.0)	6.86 ± 2.88		8.04 ± 2.70		3.33 ± 0.38		3.14 ± 1.11	
Underlying diseases	Yes	60(54.5)	7.23 ± 2.90	−1.26 (0.212)	7.82 ± 3.12	−0.65 (0.516)	3.35 ± 0.42	−0.77 (0.445)	3.27 ± 1.16	0.51 (0.609)
	No	50(45.5)	7.94 ± 2.99		8.16 ± 2.41		3.41 ± 0.33		3.38 ± 1.14	
Family history	Yes	10(9.1)	8.50 ± 268	1.06 (0.290)	7.60 ± 1.65	−0.44 (0.662)	3.30 ± 0.67	−0.45 (0.662)	3.40 ± 1.26	−0.24 (0.815)
	No	100(90.9)	7.46 ± 2.97		7.78 ± 2.99		3.39 ± 0.34		3.31 ± 1.14	
Abdominal surgery experience	Yes	22(20.0)	7.64 ± 3.37	0.15 (0.885)	7.36 ± 2.85	−1.14 (0.258)	3.25 ± 0.47	−1.88 (0.062)	3.55 ± 0.86	−1.04 (0.302)
	No	88(80.0)	7.53 ± 2.85		8.13 ± 2.79		3.42 ± 0.35		3.26 ± 1.21	
Colonoscopy experience	Yes	74(67.3)	7.97 ± 2.62	0.01 (0.052)	7.92 ± 2.89	−0.29 (0.775)	3.39 ± 0.40	0.37 (0.712)	3.39 ± 1.07	−0.96 (0.337)
	No	36(32.7)	6.69 ± 3.41		8.08 ± 2.68		3.36 ± 0.34		3.17 ± 1.30	
Meat	Preferred	33(30.0)	7.91 ± 2.84	0.36 (0.697)	7.97 ± 2.66	0.09 (0.915)	3.39 ± 0.39	0.52 (0.594)	3.58 ± 1.00	1.34 (0.266)
preference	Moderate	69(62.7)	7.38 ± 2.92		7.93 ± 2.88		3.40 ± 0.36		3.23 ± 1.18	
	Not preferred	8(7.3)	7.63 ± 3.85		8.37 ± 3.16		3.25 ± 0.57		3.00 ± 1.41	
Family	Yes	58(52.7)	7.57 ± 3.16	0.54 (0.957)	7.33 ± 3.02	−2.64 (0.009)	3.31 ± 0.44	−2.14 (0.035)	3.33 ± 1.08	−0.09 (0.928)
support	No	52(47.3)	7.54 ± 2.73		8.69 ± 2.38		3.46 ± 0.29		3.30 ± 1.23	
Constipation	Yes	28(25.5)	6.46 ± 2.76	−2.31 (0.023)	7.86 ± 2.56	−0.25 (0.802)	3.34 ± 0.40	0.68 (0.500)	3.18 ± 1.12	0.74 (0.459)
	No	82(74.5)	7.93 ± 2.93		8.01 ± 2.90		3.40 ± 0.38		3.37 ± 1.16	
Insurance	Insurance	101(91.8)	7.72 ± 2.89	2.03 (0.044)	7.92 ± 2.89	−0.98 (0.346)	3.39 ± 0.39	0.36 (0.718)	3.36 ± 1.14	−1.17 (0.244)
	Assistance	9(8.2)	5.67 ± 3.08		8.56 ± 1.74		3.34 ± 0.31		2.89 ± 1.27	

**Table 4 ijerph-19-02676-t004:** Correlations among health literacy, knowledge, compliance, and bowel cleanliness (*n* = 110).

Variables	Pearson’s Correlation Coefficients, *r* *							
	1	2	2-1	2-2	3	3-1	3-2	4
1. Health literacy	1.00						-	-
2. Knowledge of bowel preparation	0.20 *	1.00						
2-1. Knowledge of taking bowel-cleansing agents	0.36 **	0.83 **	1.00					
2-2. Dietary knowledge for colonoscopy	0.01	0.88 **	0.48 **	1.00				
3. Compliance of bowel preparation	0.29 *	0.44 **	0.39 **	0.37**	1.00			
3-1. Compliance of taking bowel-cleansing agents	0.38 **	0.28 **	0.37 **	0.13	0.65 **	1.00		
3-2. Compliance of dietary guidelines for colonoscopy	0.74	0.36 **	0.21 *	0.02	0.80 **	0.07	1.00	
4. Bowel cleanliness	0.36 **	0.04	−0.17	0.08	0.16	0.27 *	0.01	1.00

* *p* < 0.05, ** *p* < 0.001 for all the Pearson’s correlation coefficients in the table.

**Table 5 ijerph-19-02676-t005:** Difference in knowledge, compliance, and bowel cleanliness by health literacy (*n* = 110).

Variables	Categories	Health Literacy ≤ 7(*n* = 48, 43.6%)	Health Literacy > 7(*n* = 62, 56.4%)	t	p
		Mean ± SD	Mean ± SD		
Knowledgeof bowelpreparation	Total mean	7.37 ± 2.86	8.34 ± 2.70	1.989	0.049
	Knowledge of taking bowel-cleansing agents	2.43 ± 1.58	3.37 ± 1.29	3.315	0.001
	Dietary knowledge for colonoscopy	4.93 ± 1.81	5.06 ± 1.74	0.372	0.711
Compliance of bowel preparation	Total mean	3.27 ± 0.28	3.47 ± 0.42	2.950	0.004
	Compliance of taking bowel-cleansing agents	3.03 ± 0.40	3.40 ± 0.44	4.508	<0.001
	Compliance of dietary guidelines for colonoscopy	3.50 ± 0.45	3.53 ± 0.66	0.281	0.779
Bowel cleanliness	Total mean	3.06 ± 0.93	2.39 ± 1.22	3.293	0.001

The result of control variable age, gender, education level, monthly income used by ANCOVA.

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
