# Peer review of "Relationship between Health Literacy and Knowledge, Compliance with Bowel Preparation, and Bowel Cleanliness in Older Patients Undergoing Colonoscopy"

_ijerph, 2022, doi:10.3390/ijerph19052676_

Round 1

Reviewer 1 Report

  1. The abstract is well organized and contains the most relevant information. I just suggest that the authors add compliance with the ethical requirements inherent in this study.

  1. In my opinion the Introduction of the study is complete, using appropriate references. However, the authors should better explain the problem underlying this study.

  1. I suggest that the methodology section be improved and that the authors break it down into more specific subchapters. A description of some important parts is missing, namely the ethical procedures.

  1. Data analysis and results are correct require further refinement.

Results: P - recommended additions wrong p = .110; okay p = 0.110 - also in the table

  1. The discussion is well written. I suggest that the authors further develop the implications of this study for clinical practice.

  1. The list of bibliographic references is current and correct.

Author Response

I wish to thank you for kind reviews and valuable feedback on our manuscript. In response to the reviewers’ comments, we have made revisions and provided explanations in the text and table. To make it easier for you to follow how changes were made, we have made a revision-table showing our responses and/or changes to comments made by the reviewer. We hope this revision satisfactorily resolved the point that have been raised by the reviewer and thus strengthen our study. Detailed revisions are written in the attached ‘Author’ responses to the reviewer’ comments’ table.

Comment #1: The abstract is well organized and contains the most relevant information. I just suggest that the authors add compliance with the ethical requirements inherent in this study.

Authors' response:  Thank you for your valuable opinion. We completely agree with you. Based on the reviewers' comments, ‘Ethical considerations’ in the ‘Method’ part was written as a separate section. The authors have tried to comply as much as possible with ethical requirements that may arise while conducting our research.

Comment #2: In my opinion the Introduction of the study is complete, using appropriate references. However, the authors should better explain the problem underlying this study.

Authors' response:  We completely agree with you. Based on the reviewers' opinions, the correlation between the health literacy level of the elderly and the health outcomes related to colonoscopy was added and re-described.

Comment #3: I suggest that the methodology section be improved and that the authors break it down into more specific subchapters. A description of some important parts is missing, namely the ethical procedures.

Authors' response: Based on the opinions of reviewers, the methodology section has been reclassified into 5 subchapters and described. Added ‘2.4. Ethical considerations’ part.

Comment #4: Data analysis and results are correct require further refinement. Results: P - recommended additions wrong p = .110; okay p = 0.110 - also in the table.

Authors' response: Based on the reviewer's opinion, the notation of the P value in the table has been modified. Thank you for your careful review.

Comment #5: The discussion is well written. I suggest that the authors further develop the implications of this study for clinical practice.

Authors' response: We completely agree with you. Based on the reviewer's opinion, we have added the contents of the implication strategy to clinical practice in the discussion part.

Comment #6: The list of bibliographic references is current and correct.

Authors' response: Thank you for your careful review.

Reviewer 2 Report

Dr. Gwag et al. in this study using 110 human subjects demonstrated how bowel preparation and cleanliness is corelated with health literacy and knowledge. Although the sample size is not big, the main findings are understandable and vital. I don’t have major comments for this paper. I wrote my minor comments below-

  • If author can clarify when the structured questionnaire was performed? Before the colonoscopy or after? As according to title of the paper- ‘…post-colonoscopy patients’.
  • Please also provide the study protocol number in the method section.
  • Within the subjects’ female candidates were high in number but they have less health literacy, less knowledge and significantly less compliance. So could author conclude that customized education is necessary putting emphasis on female subjects. Page-6, Table-3.
  • Even though 47.3% subject are not getting family support, they have significantly high knowledge and compliance. Table- 3, page 7. How author can explain that?

Author Response

I wish to thank you for kind reviews and valuable feedback on our manuscript. In response to the reviewers’ comments, we have made revisions and provided explanations in the text and table. To make it easier for you to follow how changes were made, we have made a revision-table showing our responses and/or changes to comments made by the reviewer. We hope this revision satisfactorily resolved the point that have been raised by the reviewer and thus strengthen our study. Detailed revisions are written in the attached ‘Author’ responses to the reviewer’ comments’ table.

Comment #1:  If author can clarify when the structured questionnaire was performed? Before the colonoscopy or after? As according to title of the paper- ‘…post-colonoscopy patients’.

Authors' response: Based on the reviewers' opinions, the time when the subjects filled out the questionnaire was written in detail (2.3. Data collection). In addition, the current study title was written as 'in elderly post-colonoscopy patients', which could cause misunderstanding to readers, so it was revised to 'in older patients undergoing colonoscopy'.

Comment #2:  Please also provide the study protocol number in the method section.

Authors' response: This study was conducted as a cross-sectional descriptive research method. There is no protocol number for experimental research. In ‘Ethical consideration’ part, the approval number of the Institutional Research Ethics Review Committee has been presented.

Comment #3: Within the subjects’ female candidates were high in number but they have less health literacy, less knowledge and significantly less compliance. So could author conclude that customized education is necessary putting emphasis on female subjects. Page-6, Table-3.

Authors' response: Thank you for your valuable opinion. We completely agree with you. According to the reviewer's opinion, we re-described the need for educational intervention for older women.

Comment #4: Even though 47.3% subject are not getting family support, they have significantly high knowledge and compliance. Table- 3, page 7. How author can explain that?

Authors' response: 

Thank you for your valuable opinion.

Family support in the tool of this study did not use the commonly used concept of family support (economic, psychological, daily life support, etc.) but asked if they received help from their families only in the process of bowel preparation before colonoscopy. This means that 47.3% of the subjects of this study prepared independently, that is, on their own, without the help of their families in the process of preparing the intestine for colonoscopy. Therefore, it means that in preparing for colonoscopy, the dependence on others is not high, and the preparation process was independently understood and carried out. If the study included elderly people with physical and mental difficulties and disabilities or who had been dependent, they would have relied entirely on the support of their families or caregivers for the pre-colonoscopy preparation process. These results are considered to be due to the characteristics of the study participants without special health disorders.

Reviewer 3 Report

This is an original article concerning bowel preparation, and bowel cleanliness in 3 elderly post-colonoscopy patients 

The topic is interesting.

I have the following comments:

Moderate English-language revision needed

The introduction must be shortened avoiding the overlap with the discussion. 

Methodologically there are some limitations.
On what basis was the questionnaire designed? From who?

How were the 114 patients recruited? Consecutive? How did they receive the questionnaire?

Please consider the following articles concerning the relationship with the patient and the main topic

Novel frontiers of agents for bowel cleansing for colonoscopy. World J Gastroenterol. 2021 Dec 7;27(45):7748-7770. doi: 10.3748/wjg.v27.i45.7748

Predictive factors for inadequate bowel cleansing in colon capsule endoscopy. Gastroenterol Hepatol. 2022 Jan 19:S0210-5705(22)00006-1. English, Spanish. doi: 10.1016/j.gastrohep.2022.01.003

Internet and social media use among patients with colorectal diseases (ISMAEL): a nationwide survey. Colorectal Dis. 2020 Nov;22(11):1724-1733. doi: 10.1111/codi.15245

Please track the changes made

Author Response

I wish to thank you for kind reviews and valuable feedback on our manuscript. In response to the reviewers’ comments, we have made revisions and provided explanations in the text and table. To make it easier for you to follow how changes were made, we have made a revision-table showing our responses and/or changes to comments made by the reviewer. We hope this revision satisfactorily resolved the point that have been raised by the reviewer and thus strengthen our study. Detailed revisions are written in the attached ‘Author’ responses to the reviewer’ comments’ table.

Comment #1: Moderate English-language revision needed

Authors' response: The submitted manuscript has been edited through Editage (www.editage.co.kr), a professional English proofreading company (job code: CHOSN_2492). According to the reviewer's opinion, it has been revised once again. All revisions in the manuscript were marked in red. 

Comment #2: The introduction must be shortened avoiding the overlap with the discussion.

Authors' response: Based on the reviewer's opinion, we have shortened ‘Introduction’ part that overlaps with the discussion part.

Comment #3: Methodologically there are some limitations. On what basis was the questionnaire designed? From who?

Authors' response: Thank you for your valuable opinion. The questionnaire was selected as a tool developed for older people over 65 years of age living in Korea. The health literacy tool used the S-KHLS tool whose reliability and validity have been verified in several previous studies. Tools for knowledge and compliance of bowel preparation were also selected for the elderly in Korea. Some previous studies have been developed mainly for English-speaking populations. Our research team used these tools because they were tools related to health literacy, so it was appropriate to use a questionnaire written in Korean that was developed for the elderly in Korea.

In addition, there are various indicators used internationally for the level of bowel cleanliness, and the Aronchick bowel preparation scale, which is widely used as an indicator for quality control of colonoscopy in clinical fields in Korea, was used. The reliability and validity of the Aronchick bowel preparation scale have also been reported in a number of previous studies.

Comment #4: How were the 114 patients recruited? Consecutive? How did they receive the questionnaire?

Authors' response: Thank you for your valuable opinion. Because the hospital where this study was conducted has an endoscopy center, the number of daily visits of adult patients is high. Among the adults visiting the endoscopy center, the older people who met the selection criteria were continuously and conveniently recruited. Study participants were asked to fill out the form 30 minutes before the start of the colonoscopy. It took about 15-20 minutes to fill out the questionnaire. The questionnaire was completed in a separate area in the outpatient setting where privacy was maintained. Time and method of filling out the questionnaire were additionally described in detail in the data collection part of the research method.

Comment #5: Please consider the following articles concerning the relationship with the patient and the main topic.

Authors' response:  Thank you for your valuable opinion. After reading the references suggested by the reviewers, the necessity of newly configured internet and social media-based information platform for older people to use was additionally described in the discussion section. The newly cited reference was included in the reference list. 

Comment #6: Please track the changes made.

Authors' response:  All revisions in the manuscript were marked in red.

Reviewer 4 Report

The authors analyzed the relationship between health literacy, compliance with bowel preparation and bowel cleanliness in a sample incuding 110 elderly patients from South Korea who underwent a colonoscopy. Higher knowledge, compliance with bowel preparation and bowel cleanliness index were observed in patients with a high health literacy index.

The study is of interest and lays the basis for interventions aimed at improving health literacy among elderlies, in order to be able to conduct successful colonoscopies, which is relevant to diagnose colorectal cancer. The study is well written and clear to read. The authors also have to be commended for having included gender into the analyses, as literature to this regard is not consistent. I only have minor comments.

  • How did the authors decide which general characteristics had to be collected for included patients (for instance why was religion included?)
  • At page 3, lines 119 - 121, the authors specified: "Elderly individuals with normal cognitive levels and no special problems in daily life or communication were also included". How was this assessed?
  • I would rephrase the sentence above as to state that these individuals were "also" included might make a reader think that the sample included both individuals with and without cognition problems

Author Response

I wish to thank you for kind reviews and valuable feedback on our manuscript. In response to the reviewers’ comments, we have made revisions and provided explanations in the text and table. To make it easier for you to follow how changes were made, we have made a revision-table showing our responses and/or changes to comments made by the reviewer. We hope this revision satisfactorily resolved the point that have been raised by the reviewer and thus strengthen our study. Detailed revisions are written in the attached ‘Author’ responses to the reviewer’ comments’ table.

Comment #1: How did the authors decide which general characteristics had to be collected for included patients (for instance why was religion included?)

Authors' response: Thank you for your valuable comments. The general characteristics were selected based on the results of examining the socio-demographic variables that were mainly used in previous studies related to colonoscopy and health literacy. In the case of religion, it has been added because it has been verified in recent previous studies related to health literacy.

  • Hwang, Y.H.; Lee, G.E. The effect of health literacy and self-care performance on health care utilization of medicaid elderly. J Korean Acad Community Health Nurs 2019, 30, 484-493. https://doi.org/10.12799/jkachn.2019.30.4.484
  • Shannon, M. C.; Clement, K. G.; Steven, K. S.; Enmanuel, C.; Stacy, N. D.; Rania. A.; Chitra, R.; Ida, S.; Richard, R.; Cathy, D. M. Health literacy among medically underserved: the role of demographic factors, social influence, and religious beliefs. J Health Commun 2017, 22, 923-931. https://doi.org/10.1080/10810730.2017.1377322

In previous studies, it was reported that there was a relationship between health literacy and religion or religious belief, but in this study, religion was not a significant variable. In a follow-up study, it is also necessary to conduct a study on the difference in health literacy according to the type of religion and the level of religious activity in a specific population group.

Comment #2: At page 3, lines 119 - 121, the authors specified: "Elderly individuals with normal cognitive levels and no special problems in daily life or communication were also included". How was this assessed?

Authors' response:  Thank you for your valuable opinion. We completely agree with your opinion. The evaluation of the participants’ cognitive level, communication and daily living level was confirmed through the initial interview with the gastroenterologist at the outpatient colonoscopy room. These contents have been added and described in the maintext. In a follow-up study, we will evaluate using standardized cognitive assessment tools and ADL measurement tools. Thank you for your valuable comments.

Comment #3: I would rephrase the sentence above as to state that these individuals were "also" included might make a reader think that the sample included both individuals with and without cognition problems.

Authors' response: Thank you for your valuable opinion. I agree that it contains linguistic expressions ('also') that may be misleading to the reader. The sentence has been corrected according to the reviewer's opinion.

Round 2

Reviewer 3 Report

I'm satisfied with the changes made